# FastRPB: a Scalable Relative Positional Encoding for Long Sequence Tasks

## Abstract

Transformers achieve remarkable performance in various domains, including NLP, CV, audio processing, and graph analysis. However, they do not scale well on long sequence tasks due to their quadratic complexity w.r.t. the input's length. Linear Transformers were proposed to address this limitation. However, these models have shown weaker performance on the long sequence tasks comparing to the original one. In this paper, we explore Linear Transformer models, rethinking their two core components. Firstly, we improved Linear Transformer with **S**hift-**I**nvariant **K**ernel **F**unction **SIKF**, which achieve higher accuracy without loss in speed. Secondly, we introduce **FastRPB** which stands for **Fast R**elative **P**ositional **B**ias, which efficiently adds positional information to self-attention using Fast Fourier Transformation. FastRPB is independent of the self-attention mechanism and can be combined with an original self-attention and all its efficient variants. FastRPB has $\mathcal{O}(N \log N)$ computational complexity, requiring $\mathcal{O}(N)$ memory w.r.t. input sequence length $N$.

We compared introduced modifications with recent Linear Transformers in different settings: text classification, document retrieval, and image classification. Extensive experiments with FastRPB and SIKF demonstrate that our model significantly outperforms another efficient positional encodings method in accuracy, having up to x1.5 times higher speed and requiring up to x10 times less memory than the original Transformer.

## 1 Introduction

Transformer architecture (Vaswani et al., 2017) originally proposed for machine translation tasks has shown impressive results in a wide range of domains, including natural language processing, image recognition, audio captioning, graph analysis, and bioinformatics (Lin et al., 2021). However, in applications that require processing long sequences, the benefits of transformers are often accompanied by high consumption of computational and memory resources. The main bottleneck is the transformer's core component, the self-attention mechanism. Self-attention computes similarity scores for all pairs of tokens in the input sequence, and therefore, it has a quadratic complexity $\mathcal{O}(N^2)$ in computations and memory relative to the length of the input sequence $N$[1].

Recently, several approaches have been introduced to reduce the computational complexity and memory footprint of self-attention. Some works utilize the sparsity of the attention map (Beltagy et al., 2020), others express self-attention as a linear dot-product of kernel feature maps $\phi(\cdot)$ (Katharopoulos et al., 2020), or utilize random feature vectors (Choromanski et al., 2020). Proposed approaches reduce the computational complexity to $\mathcal{O}(N)$[2]. One of the promising variants of a transformer is the *Linear Transformer* (Katharopoulos et al., 2020) since, along with linear complexity, it requires constant $\mathcal{O}(1)$ memory in auto-regressive language modeling. Experiments with the long sequence benchmark Long Range Arena (LRA) (Tay et al., 2020)[3] have indeed shown that

---

[1]The full complexity of self-attention also depends on attention head size $D$. For the original self-attention, complexity is $\mathcal{O}(N^2 D)$

[2]In contrast, for *Linear Transformer* (Katharopoulos et al., 2020; Choromanski et al., 2020), the complexity of linear self-attention is $\mathcal{O}(ND^2)$. In long sentences, $N$ is assumed to be around thousands of tokens. Therefore, switching to linear self-attention appears beneficial.

[3]In benchmark sequences ranging from 1K to 16K tokens

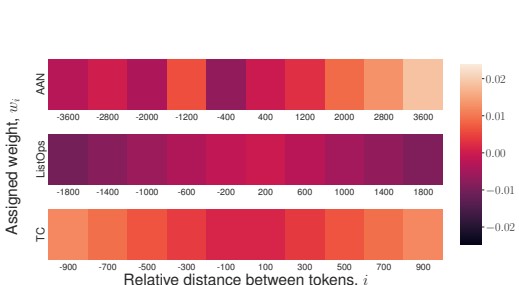 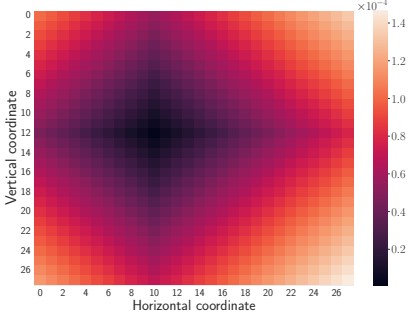

Figure 1: Learned weights $w_i$ assigned to pairwise distances between tokens $i$ in FastRPB 1D for different text LRA tasks.

Figure 2: Learned FastRPB 2D weights assigned to distances from pixel (12, 10) to each other pixel in MNIST $28 \times 28$ image classification.

the *Linear Transformer* is 5x times faster than the vanilla Transformer in training speed. However, the drawback of this architecture is lower performance compared to the original Transformer.

One way to reduce the performance gap between the *Linear Transformer* and the original one is to select a more suitable kernel function $\phi(\cdot)$ in linear attention (Choromanski et al., 2020; Schlag et al., 2021). The poor performance of efficient transformers on LRA can also be attributed to the model's ability to capture positional information. The original Transformer model utilizes only absolute positional information, which is added through positional embeddings to contextual embeddings of the tokens. Other approaches, which enrich self-attention with additional information about relative distances between tokens, have recently shown visible improvements in performance. Some of them directly add a matrix of relative distances to the attention map (Shaw et al., 2018), others compute separate attention scores between positional embeddings (He et al., 2020). We hypothesize that adding relative positional information could improve efficient transformers. However, most of the current implementations possess quadratic computational complexity, which neutralizes all efficiency of the *Linear Transformer*. To deal with this problem, a linear complexity stochastic positional encoding (SPE) was proposed (Liutkus et al., 2021). Despite linear asymptotic, SPE remains relatively inefficient in training time due to its stochastic nature, while the improvement in accuracy it brings is relatively small on several LRA tasks.

The contribution of this paper is two-fold. At first, we propose the **S**hift-**I**nvariant **K**ernel **F**unction (**SIKF**). It could be used as a kernel for the *Linear Transformer* model and holds the shift-invariance property of $\mathrm{softmax}$ in the original attention. Second, we propose **F**ast **R**elative **P**ositional **B**ias (**FastRPB**) — Fast Fourier Transform-based bias for self-attention that represents relative positional information within sequences, has $\mathcal{O}(N \log N)$ complexity and requires only $\mathcal{O}(N)$ memory. FastRPB is orthogonal to the self-attention mechanism and can be combined with both efficient and original implementations.

We observed that SIKF is comparable to more complex kernels (Choromanski et al., 2020; Schlag et al., 2021) while being as fast as the original one (Katharopoulos et al., 2020). We also evaluated FastRPB under different long-context scenarios, such as image classification and Long Range Arena tasks. Through a comprehensive study, we showed that the proposed technique outperforms the prior fast positional encoding method (Liutkus et al., 2021) by a significant margin without adding a substantial computational footprint.

## 2 RECENT WORKS

### 2.1 ATTENTION MECHANISM

The core component of Transformer (Vaswani et al., 2017) is the attention layer, which computes attention weights $A_{m,n}$, measuring how important the role of $n$-th key word is in shaping the meaning of $m$-th output word. Using $A_{m,n}$ we can construct attention matrix $\boldsymbol{A} \in \mathbb{R}^{M \times N}$, and rewrite

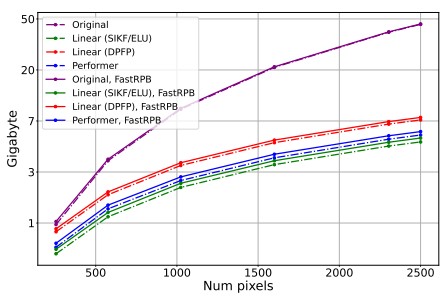

(a) Evaluation memory consumption.

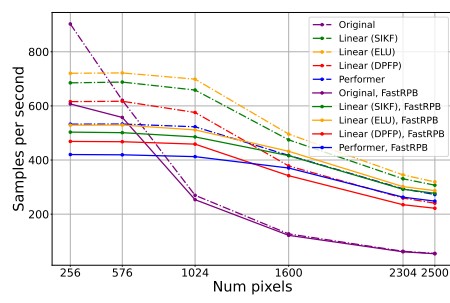

(b) Evaluation time.

Figure 3: Evaluation time and memory for various type of transformer on Nvidia A100 with respect to number of pixels in the input image. For memory consumption y-axis is log-scaled.

the equation using matrix notation. The output of the attention layer $\boldsymbol{Y}$ is defined based on three matrices $\boldsymbol{Q} \in \mathbb{R}^{M \times D}$, $\boldsymbol{K} \in \mathbb{R}^{N \times D}$ and $\boldsymbol{V} \in \mathbb{R}^{N \times D}$ (Queries, Keys, and Values) as follows:

$$\boldsymbol{Y} = \boldsymbol{AV} = \text{softmax}(\boldsymbol{\mathcal{A}})\boldsymbol{V} = \text{softmax}(\boldsymbol{QK}^T/\sqrt{D})\boldsymbol{V} \tag{1}$$

In vanilla Transformer, the attention matrix $\boldsymbol{A}$ is computed explicitly, which leads to a $\mathcal{O}(MND)$ complexity, and $\mathcal{O}(MN)$ memory to store the matrix[4].

## 2.2 Efficient Attention Mechanism

*Linear Transformer* variants (Katharopoulos et al., 2020; Choromanski et al., 2020) are a way to reduce the complexity of attention from quadratic to linear using the associative property of matrix products and kernel reformulation of attention.

By substituting the $\text{softmax}$ function in Equation 1, we obtain $m$-th row $\boldsymbol{y}_m$ of matrix $\boldsymbol{Y}$:

$$\boldsymbol{y}_m = \frac{\sum_n \exp(\boldsymbol{q}_m^T \boldsymbol{k}_n/\sqrt{D})\boldsymbol{v}_n}{\sum_n \exp(\boldsymbol{q}_m^T \boldsymbol{k}_n/\sqrt{D})} = \frac{\sum_n \text{sim}(\boldsymbol{q}_m, \boldsymbol{k}_n)\boldsymbol{v}_n}{\sum_n \text{sim}(\boldsymbol{q}_m, \boldsymbol{k}_n)} \tag{2}$$

where $\exp(\boldsymbol{q}_m^T \boldsymbol{k}_n/\sqrt{D})$ is generalized by any arbitrary defined similarity function $\text{sim}(\boldsymbol{q}_m, \boldsymbol{k}_n)$.

The core idea of *Linear Transformer* is to replace $\text{sim}(\boldsymbol{q}_m, \boldsymbol{k}_n)$ with a dot-product using kernel function $\phi(\cdot)$ and then use an associative property of matrix products as follows:

$$\boldsymbol{y}_m = \frac{\sum_n \phi(\boldsymbol{q}_m)^T \phi(\boldsymbol{k}_n)\boldsymbol{v}_n}{\sum_n \phi(\boldsymbol{q}_m)^T \phi(\boldsymbol{k}_n)} = \frac{\phi(\boldsymbol{q}_m)^T \sum_n \phi(\boldsymbol{k}_n)\boldsymbol{v}_n}{\phi(\boldsymbol{q}_m)^T \sum_n \phi(\boldsymbol{k}_n)} \tag{3}$$

Original attention has $\mathcal{O}(N^2 D)$ time complexity, where $N$ represents the sequence length, and $\mathcal{O}(N^2)$ for the memory footprint, while linear attention has time and memory complexity $\mathcal{O}(ND^2)$, which scales linearly with sequence length $N$.

## 2.3 Kernel function variants

An open question is selecting an appropriate kernel function for *Linear Transformer* since different kernel functions dramatically affect trained model accuracy and speed.

**ELU + 1**. Originaly proposed kernel is an element-wise $\text{ELU}(\cdot) + 1$ (Katharopoulos et al., 2020):

$$\phi(x) = \text{ELU}(x) + 1 = \begin{cases} x + 1, & x > 0 \\ \exp(x), & x \leq 0 \end{cases} \tag{4}$$

The choice of $\text{ELU}(\cdot) + 1$ over $\text{ReLU}(\cdot)$ was prompted by its non-zero gradients for negative values.

---

[4]In case of self-attention, $M$ equals to $N$, and thus the complexity is $\mathcal{O}(N^2 D)$ and memory requirements is $\mathcal{O}(N^2)$.

**Performer**. The core idea is to approximate the $\mathrm{softmax}$ on average using random features (Choromanski et al., 2020). The kernel function is evaluated as:

$$\phi(\boldsymbol{x}) = \frac{h(\boldsymbol{x})}{\sqrt{m}} \begin{bmatrix} \exp(\mathbf{R}\boldsymbol{x}) \\ \exp(-\mathbf{R}\boldsymbol{x}) \end{bmatrix}, \text{ where } h(\boldsymbol{x}) = \frac{1}{\sqrt{2}} \exp\left(-\frac{1}{2}\|\boldsymbol{x}\|\right) \tag{5}$$

Here $\begin{bmatrix} \exp(\mathbf{R}\boldsymbol{x}) \\ \exp(-\mathbf{R}\boldsymbol{x}) \end{bmatrix}$ states for concatenation of vectors $\exp(\mathbf{R}\boldsymbol{x})$ and $\exp(-\mathbf{R}\boldsymbol{x})$ along feature dimension, each row $\mathbf{r} \in \mathbb{R}^D$ of matrix $\mathbf{R} \in \mathbb{R}^{R \times D}$ is sampled from normal distribution $\mathcal{N}(\mathbf{0}, \boldsymbol{I}_D)$, and dimension size $R$ is a hyperparameter.

The main drawback of Performer is that sampling of matrix $\mathbf{R}$ requires extra computations and introduces variance into the model's output.

**DPFP**. Deterministic parameter-free projection is an alternative approach (Schlag et al., 2021). The kernel function, designed to facilitate orthogonality in the projected space $\mathbb{R}^{D_{\mathrm{proj}}}$, is described as:

$$\phi_{i \cdot \nu}(\boldsymbol{x}) = \mathrm{ReLU}\left(\begin{bmatrix} \boldsymbol{x} \\ -\boldsymbol{x} \end{bmatrix}\right)_i \mathrm{ReLU}\left(\begin{bmatrix} \boldsymbol{x} \\ -\boldsymbol{x} \end{bmatrix}\right)_{i+\nu}, \text{ where } \phi : \mathbb{R}^D \to \mathbb{R}^{D_{\mathrm{proj}}} \tag{6}$$

here $i \cdot \nu$ indicates the index of vector $\phi(\boldsymbol{x})$, $i \in \{1, 2, ..., 2D\}$ is an index and $\nu \in \{1, 2, ..., 2D-1\}$ is a hyperparameter, controlling the capacity of kernel function $\phi(\cdot)$. *Linear Transformer* with DPFP model outperforms model with default kernel and Performer, even if $D_{\mathrm{proj}}$ is relatively small. Also, DPFP showed to be faster than models utilizing random features, but still slightly slower than $\mathrm{ELU} + 1$.

## 2.4 Positional Information

Attention is permutation-invariant, which means that the attention layer does not make use of the order of the sequence. There exist different ways to encode positional information in the attention:

**Absolute Positional Encoding (APE)** proposed in original Transformer architecture uses real-valued vector $\boldsymbol{p}_i \in \mathbb{R}^D$ assigned to each positions $i$. Some approaches, such as vanilla Transformer (Vaswani et al., 2017), use predefined vectors, while others employ learnable vectors, e.g., in BERT (Devlin et al., 2018).

**Relative Positional Encoding (RPE)** is complement to the absolute positional encoding, which explicitly adds relative positional information between vectors (Shaw et al., 2018) to the model. Raffel et al. (2019) proposed to directly embed positional information to the matrix $\boldsymbol{\mathcal{A}}$ (see the Equation 1). This approach was then improved by separating semantic correlation of words and their positions correlation by Ke et al. (2020). The component $\mathcal{A}_{m,n}$ of matrix $\boldsymbol{\mathcal{A}}$ then calculated as:

$$\mathcal{A}_{m,n} = \frac{1}{\sqrt{D}} \boldsymbol{q}_m^T \boldsymbol{k}_n + \frac{1}{\sqrt{D}} (\boldsymbol{U}_Q \boldsymbol{p}_m)^T (\boldsymbol{U}_K \boldsymbol{p}_n) \tag{7}$$

where $\boldsymbol{p}_n$ and $\boldsymbol{p}_m$ are embeddings of corresponding positions $n$ and $m$, and $\boldsymbol{U}_Q, \boldsymbol{U}_K \in \mathbb{R}^{D \times D}$ are learnable projection matrices for the positional embedding.

By design, these approaches have quadratic computational complexity. Thus their usage with *Linear Transformer* is challenging since the naive application will neutralize all effectiveness of linear computation time.

**Stochastic Positional Encoding (SPE)** proposed by Liutkus et al. (2021) is, to the best of our knowledge, currently the only positional encoding method compatible with *Linear Transformer* variants due to its linear complexity. The key idea for SPE is to represent the attention relative distances matrix as a covariance. Following the notation from equation 1, we can express $\mathcal{A}_{m,n}$ as:

$$\mathcal{A}_{m,n} = \sum_{d=1}^{D} Q_{m,d} \cdot \mathcal{P}_d(m,n) \cdot K_{n,d} / \sqrt{D}, \text{ where } \mathcal{P}_d(m,n) = \mathbb{E}\left[\overline{\mathrm{q}}_d(m) \cdot \overline{\mathrm{k}}_d(n)\right] \tag{8}$$

here $Q_{m,d}$ and $K_{n,d}$ are components of matrices $\boldsymbol{Q}$ and $\boldsymbol{K}$ respectively. $\overline{\mathrm{q}}_d(m)$ and $\overline{\mathrm{k}}_d(n)$ are two real and zero-mean random variables such that their covariance function matches $\mathcal{P}_d$. Varying the structure of matrices $\mathcal{P}_d$ authors designed two variants of SPE: sinSPE and convSPE. The first

| | | Positional Encoding | | | | |
|---|---|---|---|---|---|---|
| | | None | FastRPB | sinSPE | convSPE | RPE |
| **AAN** | Original | OOM | OOM | OOM | OOM | OOM |
| | Linear, DPFP | 61.01 ± 0.79 | 64.79 ± 1.52 | 61.53 ± 0.75 | 63.52 ± 0.71 | N/A |
| | Linear, SIKF | 59.51 ± 0.3 | **67.19 ± 1.64** | 62.0 ± 0.36 | 58.93 ± 1.65 | N/A |
| | Linear, ReLU | 58.78 ± 0.93 | 64.94 ± 1.6 | 62.39 ± 0.59 | 61.00 ± 1.34 | N/A |
| | Performer | 59.84 ± 1.46 | 66.65 ± 0.91 | 60.00 ± 1.20 | 57.22 | N/A |
| **ListOps** | Original | 14.43 ± 4.73 | 14.6 ± 4.14 | – | – | OOM |
| | Linear, DPFP | **20.67 ± 3.95** | 17.97 ± 11.68 | 17.57 ± 0.18 | 16.17 ± 5.89 | N/A |
| | Linear, SIKF | 12.55 ± 3.8 | 11.47 ± 4.79 | 15.25 ± 8.97 | 17.8 ± 0.0 | N/A |
| | Linear, ReLU | 17.58 ± 1.01 | 17.67 ± 0.59 | 17.80 ± 0.00 | 9.50 ± 1.17 | N/A |
| | Performer | 17.80 ± 0.00 | 17.75 ± 0.39 | 17.43 ± 0.32 | 17.80 | N/A |
| **CIFAR** | Original | 41.88 ± 0.48 | 39.02 ± 0.22 | – | – | N/A |
| | Linear, DPFP | 41.79 ± 0.27 | 38.73 ± 0.09 | 41.97 ± 1.24 | 41.33 ± 0.84 | N/A |
| | Linear, SIKF | 41.96 ± 0.47 | 38.89 ± 0.15 | 40.73 ± 0.58 | **42.94 ± 0.51** | N/A |
| | Linear, ReLU | 42.25 ± 0.01 | 38.44 ± 0.38 | 41.21 ± 1.18 | 39.96 ± 1.31 | N/A |
| | Performer | 41.81 ± 1.16 | 32.26 ± 9.53 | 41.12 ± 1.70 | 40.06 | N/A |
| **TC** | Original | 62.27 ± 0.8 | 62.02 ± 2.02 | – | – | 55.7 ± 1.94 |
| | Linear, DPFP | 62.78 ± 0.48 | 63.05 ± 0.62 | 62.76 ± 0.21 | 62.78 ± 0.48 | N/A |
| | Linear, SIKF | 61.64 ± 0.82 | 62.35 ± 0.24 | 63.37 ± 1.4 | 62.24 ± 0.56 | N/A |
| | Linear, ReLU | 58.78 ± 0.93 | **63.95 ± 0.16** | 62.39 ± 0.59 | 61.00 ± 1.34 | N/A |
| | Performer | 59.84 ± 1.46 | 62.66 ± 0.11 | 60.00 ± 1.20 | 57.22 | N/A |

Table 1: Experiments on Long Range Arena benchmark, the best model is boldface, the double underline is a top-2 result. Results for Performer and *Linear Transformer* (ReLU) are copied from SPE (Liutkus et al., 2021), except experiments with FastRPB. We mark experiments that failed due to memory limitations as OOM (Out of Memory). Since RPE is compatible only with the Original Transformer, we marked other experiments as Not Applicable (N/A). RPE is N/A for CIFAR since plain RPE is designed for 1D sequences. We marked experiments that were too long to train as "–".

one yields periodic covariance functions, which showed to be beneficial in such tasks as music generation. The second utilizes vanishing covariance functions, a promising concept introduced in Wang et al. (2020), which yields notably smaller validation losses in some SPE experiments.

Although SPE was beneficial in some music generation tasks, it still requires many computations due to its stochastic nature. In practice, it could be dozens of times slower than the original Transformer, as we will show feather.

## 3 APPROACH

### 3.1 SHIFT-INVARIANT KERNEL FUNCTION (SIKF)

We hypothesize that the shift-invariance property of $\mathrm{softmax}$ function (i.e., the fact that $\mathrm{softmax}_i(\boldsymbol{x} + c) = \mathrm{softmax}_i(\boldsymbol{x})$, where $\boldsymbol{x}$ is some vector, $c$ is a constant, which is added to every component of $\boldsymbol{x}$) is an important property which makes original Transformer perform better than *Linear Transformer* with an arbitrary kernel. Based on this assumption, we propose SIKF as $\phi(x) = \exp(x)$, which satisfies the property of shift-invariance. If we substitute this function in the linear attention from Equation 3, then for every real-valued constants $c$ and $d$ we will get:

$$\frac{\phi(\boldsymbol{q}_m + c)^T \sum_n \phi(\boldsymbol{k}_n + d)\boldsymbol{v}_n}{\phi(\boldsymbol{q}_m + c)^T \sum_m \phi(\boldsymbol{k}_n + d)} = \frac{e^c \phi(\boldsymbol{q}_m)^T \sum_n e^d \phi(\boldsymbol{k}_n)\boldsymbol{v}_n}{e^c \phi(\boldsymbol{q}_m)^T \sum_n e^d \phi(\boldsymbol{k}_n)} = \frac{\phi(\boldsymbol{q}_m)^T \sum_n \phi(\boldsymbol{k}_n)\boldsymbol{v}_n}{\phi(\boldsymbol{q}_m)^T \sum_n \phi(\boldsymbol{k}_n)} \quad (9)$$

Thus, attention in *Linear Transformer* with $\exp(\cdot)$ kernel function holds the same shift-invariance property as plain $\mathrm{softmax}$.

Based on our experiments, we conclude that SIKF is faster than Performer and DPFP, simultaneously has comparable accuracy, and does not provide an extra memory footprint, which is essential for scaling *Linear Transformer* on extremely long sequences.

| Model | w/o FastRPB | w/ FastRPB |
|---|---|---|
| Original | 97.34 ± 0.23 | **98.27 ± 0.19** |
| Linear, DPFP | 97.09 ± 0.19 | 97.66 ± 0.22 |
| Linear, SIKF | 96.49 ± 0.20 | 97.37 ± 0.35 |
| Linear, ELU + 1 | 94.01 ± 0.31 | 96.71 ± 0.40 |
| Performer | 96.6 ± 0.29 | 97.52 ± 0.26 |

Table 2: MNIST F1 score. All the experiments run on 4 Nvidia Tesla T4.

## 3.2 FAST RELATIVE POSITIONAL BIAS (FASTRPB)

Although adding positional information in the attention mechanism is beneficial for model accuracy, current approaches are relatively inefficient in long sequences in terms of speed and memory footprint. In this context, there is a desire to design an approach that will add relative positional information to attention efficiently and simultaneously be compatible with various efficient attention modifications. To achieve this goal, we propose FastRPB[5] as a separate term for attention.

More formally, output matrix $\boldsymbol{Y}$ of attention layer with FastRPB is defined as:

$$\boldsymbol{Y} = \text{AttentionVariant}(\boldsymbol{Q}, \boldsymbol{K}, \boldsymbol{V}) + \boldsymbol{W}\boldsymbol{V} \tag{10}$$

here matrix $\boldsymbol{W} \in \mathbb{R}^{M \times N}$ consists of learnable weights $W_{m,n}$ representing relative distances between $m$ and $n$ embedding vectors from matrix $\boldsymbol{V}$. Note that Equation 10 is invariant of choosing a specific attention mechanism and could be used with both vanilla attention and its linear variants (Equations 1 and 3 respectively).

One can think of the matrix $\boldsymbol{W}$ as a bias term to the usual attention matrix $\boldsymbol{A}$ from equation 1, correcting the attention weights according to the relative distance between corresponding tokens. However, adding positional bias term in the Equation 10 still requires $\mathcal{O}(NMD)$ computations due to the matrix product and $\mathcal{O}(NM)$ memory to store the bias matrix $\boldsymbol{W}$[6]. In this regard, in the following two subsections, we will construct FastRPB positional bias terms matrices $\boldsymbol{W}_{1d}$ and $\boldsymbol{W}_{2d}$ for different types of sequences, that can be efficiently multiplied by $\boldsymbol{V}$. $\boldsymbol{W}_{1d}$ utilized in the case of 1D sequences (e.g., natural language texts), and its coefficients will correspond to distances between words in 1D sequences. For 2D sequences we will utilize $\boldsymbol{W}_{2d}$, which coefficients will represent distances between elements of 2D sequences (i.e., pixels). We will show that these specific matrices $\boldsymbol{W}_{1d}$ and $\boldsymbol{W}_{2d}$ could be multiplied with $\boldsymbol{V}$ using only $\mathcal{O}(DN \log N)$ computations, and requiring only $\mathcal{O}(N)$ memory.

Further, we work with self-attention — a variant of attention mechanism, where input and output sequences lengths are the same, i.e. $N = M$. In the general case of attention, when we have an input sequence of length $N$ and an output sequence of length $M$, we can pad the longer one to make the input and output lengths match.

### 3.2.1 1D SEQUENCE CASE

Suppose we have a 1D sequence with $N$ tokens. In such a sequence, there are exactly $2N - 1$ relative distances between tokens[7]. Let's assign a learnable parameter $w_i \in \mathbb{R}$ for each relative distance $i \in \{-N + 1, ..., -1, 0, 1, ..., N - 1\}$. We then will obtain a set of parameters:

$$\{w_{-N+1}, ..., w_{-1}, w_0, w_1, ..., w_{N-1}\} \tag{11}$$

Next we will construct matrix $\boldsymbol{W}_{1d}$ using parameters $\{w_i\}_{i=-N+1}^{N-1}$. The basic intuition is to make $(n, m)$-th element of matrix $\boldsymbol{W}_{1d}$ to be assigned to the relative distance between from $m$-th token to

---

[5]There was a desire to name FastRPB as FastRPE to represent that it is like a faster RPE. However, we change one letter to emphasize that FastRPB is orthogonal to the selection of an attention algorithm and could be seen as a separate bias term to the attention map.

[6]$\mathcal{O}(N^2D)$ and $\mathcal{O}(N^2)$ respectively in the case of self-attention

[7]Relative distance from $m$-th token to $n$-th token is $m - n$, which can have both positive and negative values, in this regard we have exactly $2N - 1$ learnable parameters

$n$-th token, i.e., $w_{m-n}$. Therefore matrix $\boldsymbol{W}_{1d}$ will have the following structure:

$$\boldsymbol{W}_{1d} = \begin{pmatrix} w_0 & w_1 & w_2 & \cdots & w_{N-1} \\ w_{-1} & w_0 & w_1 & \cdots & w_{N-2} \\ \vdots & \vdots & \vdots & \ddots & \\ w_{-N+1} & w_{-N+2} & w_{-N+3} & \cdots & w_0 \end{pmatrix} \tag{12}$$

By definition, $\boldsymbol{W}_{1d}$ is a Toeplitz matrix (Gray, 2001). A naive way to calculate product $\boldsymbol{W}_{1d} \cdot \boldsymbol{V}$ requires $O(N^2 D)$ computations in case of self-attention[8]. It turns out that it can be efficiently multiplied by a matrix $\boldsymbol{V}$ according to the following proposition:

**Proposition 3.1** *For every Toeplitz matrix $\boldsymbol{W}_{1d} \in \mathbb{R}^{N \times N}$ and for every matrix $\boldsymbol{V} \in \mathbb{R}^{N \times D}$, matrix product $\boldsymbol{W}_{1d} \cdot \boldsymbol{V}$ requires $\mathcal{O}(DN \log N)$ operations and $\mathcal{O}(N)$ memory. Here $N$ is length of the input sequence.*

Using proposition 3.1 we can claim that FastRPB for 1D sequence will require $\mathcal{O}(DN \log N)$ computational operations. Moreover, we only need $\mathcal{O}(N)$ memory for storing parameters $\{w_i\}_{i=-N+1}^{N-1}$, generating Toeplitz matrix $\boldsymbol{W}_{1d}$. For the proof and a more detailed explanation of proposed properties see Appendix B.2.

### 3.2.2 2D SEQUENCE CASE

In the case of 2D sequences (e.g., images), a similar matrix to $\boldsymbol{W}_{1d}$ could be defined. We will call this matrix $\boldsymbol{W}_{2d}$, and it will consist of learnable weights assigned to pairwise distances from each pixel of the image to the rest of the pixels. Here we will consider only the case of square images of size $N \times N$ [9].

The natural way to process images of the size $N \times N$ in the Transformer model is to flatten them into a vector of size $N^2$. In this regard, matrix of pairwise distances $\boldsymbol{W}_{2d}$ needs to be of size $N^2 \times N^2$. For simplicity we will assume images to be presented as a $N \times N$ matrix, and $\boldsymbol{W}_{2d}$ will be expressed as a tensor $\mathbf{W}_{2d}$ of size $(N \times N) \times (N \times N)$, which $(n, m, l, k)$ component represents distance from pixel $(l, k)$ to pixel $(n, m)$.

We will assume the distance between two pixels to be a sum of two distances: the vertical and horizontal[10]. In this regard, tensor $\mathbf{W}_{2d}$ can be decomposed on vertical and horizontal tensor terms $\mathbf{X}$ and $\mathbf{Y}$, respectively, as $\mathbf{W}_{2d} = \mathbf{X} + \mathbf{Y}$. Similar to the 1D case, we will assign shared learnable parameters $\{w_i\}_{i=-N+1}^{N-1}$ for horizontal and vertical distances. Then to compute a matrix product of tensor $\boldsymbol{W}_{2d}$ of size $N^2 \times N^2$ with matrix $\boldsymbol{V}$ of size $N^2 \times D$ we will simply flatten tensors $\mathbf{X}$ and $\mathbf{Y}$, and obtain matrices $\boldsymbol{X}_{\text{flat}}$ and $\boldsymbol{Y}_{\text{flat}}$ of shape $N^2 \times N^2$, and compute $\boldsymbol{W}_{2d}\boldsymbol{V}$ as $\boldsymbol{X}_{\text{flat}}\boldsymbol{V} + \boldsymbol{Y}_{\text{flat}}\boldsymbol{V}$.

It turns out that the structure of matrices $\boldsymbol{X}_{flat}$ and $\boldsymbol{Y}_{flat}$ is very similar to Toeplitz matrices from the Section 3.2.1. In this regard, $\boldsymbol{W}_{2d}$ can be multiplied by $\boldsymbol{V}$ efficiently, according to the following proposition:

**Proposition 3.2** *Product of matrix $\boldsymbol{W}_{2d} \in \mathbb{R}^{N^2 \times N^2}$ with matrix $\boldsymbol{V} \in \mathbb{R}^{N^2 \times D}$ using matrices $\boldsymbol{X}_{flat}$ and $\boldsymbol{Y}_{flat}$ requires $\mathcal{O}(DN \log N)$ and $\mathcal{O}(N)$ memory.*

Using the above proposition 3.2 we conclude that FastRPB 2D will require $\mathcal{O}(DN \log N)$ computational operations. Moreover, it is not essential to store whole tensors $\mathbf{X}$ and $\mathbf{Y}$ to compute the product, we only need $\mathcal{O}(N)$ of memory for parameters $\{w_i\}_{i=-N+1}^{N-1}$ generating this tensors. (See Appendix B.3 for the proof).

## 4 EXPERIMENTS

**Long Range Arena**. We evaluate proposed methods in the Long Range Arena (Tay et al., 2020), a benchmark for efficient Transformers with several text and image long sequence tasks. The

---

[8]Matrix $\boldsymbol{V}$ has size $N \times D$, where $D$ is a hidden size

[9]If we work with non-square images of size $N \times M$, we can simply pad them with zeros to make it square.

[10]If we consider two pixels $p_1 = (3, 2)$ and $p_2 = (0, 1)$ of image of size $4 \times 4$, the horizontal relative distance from pixel $p_2$ to pixel $p_1$ then will be $2 - 1$, and the vertical will be $3 - 0$

| | | Training time (hours) | | | | | Peak Memory Usage (GB) | | | | |
|---|---|---|---|---|---|---|---|---|---|---|---|
| | | None | FastRPB | sinSPE | convSPE | RPE | None | FastRPB | sinSPE | convSPE | RPE |
| **AAN** | Original | OOM | OOM | OOM | OOM | OOM | 5.81 | 6.03 | – | – | 9.69 |
| | Linear, DPFP | **0.36** | 0.43 | 2.45 | 15.41 | N/A | **0.31** | 0.57 | 0.78 | 0.87 | N/A |
| | Linear, SIKF | **0.36** | 0.45 | 1.22 | 9.12 | N/A | **0.31** | 0.57 | 0.78 | 0.83 | N/A |
| | Linear, ReLU | **0.36** | 0.45 | 1.26 | 9.13 | N/A | **0.31** | 0.57 | 0.78 | 0.83 | N/A |
| | Performer | 0.6 | 0.79 | 1.6 | 10.52 | N/A | 0.54 | 0.68 | 0.77 | 0.87 | N/A |
| **ListOps** | Original | 0.74 | 0.85 | – | – | OOM | 3.25 | 3.49 | – | – | 3.66 |
| | Linear, DPFP | 0.26 | 0.36 | 2.3 | 13.8 | N/A | 0.68 | 0.85 | 1.32 | 1.33 | N/A |
| | Linear, SIKF | **0.24** | 0.34 | 0.95 | 6.85 | N/A | 0.68 | 0.85 | 1.32 | 1.33 | N/A |
| | Linear, ReLU | **0.24** | 0.34 | 0.98 | 6.87 | N/A | 0.68 | 0.85 | 1.32 | 1.33 | N/A |
| | Performer | 0.38 | 0.48 | 1.06 | 8.7 | N/A | **0.67** | 0.9 | 1.03 | 1.32 | N/A |
| **CIFAR** | Original | **1.94** | 1.97 | – | – | N/A | **12.36** | 12.39 | – | – | N/A |
| | Linear, DPFP | **1.94** | 1.97 | 2.07 | 2.44 | N/A | **12.36** | 12.39 | 12.57 | 12.58 | N/A |
| | Linear, SIKF | **1.94** | 1.96 | 2.06 | 2.43 | N/A | **12.36** | 12.39 | 12.57 | 12.58 | N/A |
| | Linear, ReLU | **1.94** | 1.97 | 2.06 | 2.44 | N/A | **12.36** | 12.39 | 12.57 | 12.58 | N/A |
| | Performer | **1.94** | 1.96 | 2.07 | 2.44 | N/A | **12.36** | 12.39 | 12.57 | 12.58 | N/A |
| **TC** | Original | 1.81 | 2.24 | – | – | 6.58 | 0.52 | 0.52 | – | – | 0.78 |
| | Linear, DPFP | 1.48 | 1.58 | 4.25 | 13.56 | N/A | **0.23** | **0.23** | 0.33 | 0.43 | N/A |
| | Linear, SIKF | 1.56 | 1.62 | 3.52 | 9.85 | N/A | **0.23** | **0.23** | 0.33 | 0.43 | N/A |
| | Linear, ReLU | **1.46** | 1.62 | 3.54 | 9.85 | N/A | **0.23** | **0.23** | 0.33 | 0.43 | N/A |
| | Performer | 1.93 | 2.38 | 5.81 | 40.09 | N/A | 0.33 | 0.41 | 0.33 | 0.47 | N/A |

Table 3: Benchmark results on LRA with experiment setup proposed in SPE (Liutkus et al., 2021). All the above experiments were conducted using a single Nvidia A100 GPU. The best model is boldface, the double underline is a top-2 result, and the single underline is a top-3 result. We mark experiments that failed due to limited memory as OOM (Out of Memory). We did not run Original Transformer with sinSPE and convSPE since they require too much time to train.

main challenge of these tasks is dictated by the large sequence lengths, which average varies from $1K$ to $16K$ token[11]. In our experiments, we used the following tasks from this benchmark: (1) ListOps, which test if a model is capable of parsing hierarchical expressions (Nangia & Bowman, 2018); (2) TC, which consist of movie review sentiment analysis on the IMDB corpus (Maas et al., 2011); (3) All About NLP (AAN), which evaluates the model performance in matching and retrieval tasks (Radev et al., 2013); and (4) CIFAR10, image classification dataset (Krizhevsky, 2009).

We compared Vanilla Transformer, *Linear Transformer* with all kernels observed in section 2.3 with SIKF, combined with different positional encodings, namely: sineSPE, convSPE, FastRPB. We also report results of experiments w/o adding any relative positional information. All models also used trainable Absolute Positional Encodings.

All experiments and hyperparameters were conducted following instructions for the LRA dataset. We also used LRA tasks to measure memory during the evaluation and computational footprints during the training.

**MNIST**. Due to the fact that in LRA CIFAR10 experiment only single layer transformer is used, we conducted another image recognition experiment with larger networks. We evaluated all the above models w/ and w/o FastRPB on classical image classification dataset MNIST (Lecun et al., 1998). In this experiment, we did not compare FastPRE with other positional encoding methods since, as we observed in LRA, they require dozens of times more training time in experiments with multi-layer transformers with large size of hidden states.

For all experiments, we used a model with 8 layers, 8 attention heads, a hidden size equal to 256, and batch equal to 160. We trained models using AdamW optimizer and made 20 runs of Bayesian hyperparameter search to find the optimal learning rate and than trained all models for 25 epochs. Parameters are presented in Appendix A. We linearly decayed the learning rate to 0 during the training. Final results are averaged over 10 runs with different values of random seed.

---

[11]We do not include another synthetic image classification task, Pathfinder, since we were unable to reproduce the results, obtained in the original paper (Tay et al., 2020).

## 5 RESULTS

**Long Range Arena**. See Table 1 for the evaluation results. *Linear Transformer* with SIKF kernel comes out in the top two results for every dataset, except CIFAR10, which we will discuss separately. Memory footprint (see Table 3) of SIKF is indeed very tight to the ReLU, DPFP, and Performer, while DPFP and Performer constantly performed slower (up to **1.4x** times).

*Linear Transformer* equipped with FastRPB showed significantly higher results on ANN and TC both in memory, speed, and accuracy, achieving even better results than Original Transformer. More-over, architectures with FastRPB, confirming the above propositions 3.1 and 3.2, proved to have memory and computation consumption comparable with the default architectures. For the ListOps dataset, the best performance was obtained by the model without any relative positional encoding. We attribute this result to the fact that relative distances can be confusing in parse hierarchical struc-ture, such as expressions for ListOps or source code (e.g., the distance between IF and ELSE in source code can be pretty large, however, these statements are inwardly connected). We measured memory and computational footprint of models (see Table 3), according to which FastRPB requires up to **30x** less time to train than convSPE, and up to **3x** less then sinSPE. Simultaneously, FastRPB has **1.5x** smaller memory footprint on evaluation then sinSPE and convSPE. Thus we conclude that FastRPB is the fastest and the most accurate method among others.

The learned weights of FastRPB for different text tasks sequence is presented in Figure 1, where $x$-axis represents relative distance $m-n$ from $n$-th token to $m$-th token, and color denotes the value of $(W_{1d})_{n,m} = w_{m-n}$. In the AAN task FastRPB forces the model to attend more to the very end of the text. Such observation can be attributed to the fact that AAN mainly consists of scientific texts, which final part usually contains a conclusion. As for ListOps, learned FastRPB weights are usually relatively small, which supports the hypothesis that relative positional encodings in such tasks should be designed using the text hierarchy information. In the TC task, learned weights mainly draw the model's focus forward and backward to enable the model to capture long-range dependencies.

In experiments with CIFAR, usage of FastRPB was shown to decrease the model's performance, and the best result was obtained with convSPE, which outperformed others by a significant margin. We attributed this to the experiment setup, where a single-layer network is utilized. In this regard, we conduct the experiments with a more extensive network on the MNIST dataset.

**MNIST**. In this task, each of the above models equipped with FastRPB showed superior perfor-mance (see Table 2), requiring quite a small amount of additional memory and computational time (see Figures 3a and 3b respectively). As can be observed from the plots, of *Linear Transformer* with FastRPB is up to **10x** times less than the original ones. Moreover, in terms of speed *Linear Transformer* with FastRPB is **5x** time faster in evaluation time compared to original Transformer.

The detailed view of trained FastRPB could be found in Figure 2, where we learned slice $(\mathbf{W}_{2d})_{:,:,12,10}$, which represents pairwise distances from pixel $(12, 10)$ to every other pixel of $28 \times 28$ MNIST image. We observed that FastRPB enforced the model to look at more distinct pixels rather than to close ones.

## 6 CONCLUSION

We presented two novel approaches aimed to increase the accuracy of the *Linear Transformer* model without additional memory footprint and significant loss in speed. Contribution is two-fold: we first make linear attention shift-invariance and then add a bias term to attention scores, representing pairwise distances between tokens of the sequence. We computed this bias term efficiently and achieved $\mathcal{O}(N \log N)$ complexity and $\mathcal{O}(N)$ memory w.r.t. sequence length.

We demonstrate the superiority of our approach among others using four long sequence tasks from the Long Range Arena benchmark and on the MNIST dataset. Our model performs significantly better than previous approaches, obtaining the best accuracy on several tasks while being almost as efficient in terms of speed and memory as plain *Linear Transformer*.

We believe that the principles presented in this work can serve as a basis for future research on the role of positional information encoding in transformer architectures. To this end, we open-source all the code and trained models.

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

## A   MNIST HYPERPARAMTERTS SEARCH

| Model | w/o FastRPB | w/ FastRPB |
|---|---|---|
| Original | $1.05 \cdot 10^{-4}$ | $1.35 \cdot 10^{-4}$ |
| Linear, DPFP | $1.3 \cdot 10^{-4}$ | $0.7 \cdot 10^{-4}$ |
| Linear, SIKF | $1.25 \cdot 10^{-4}$ | $1.45 \cdot 10^{-4}$ |
| Linear, ELU + 1 | $1.3 \cdot 10^{-4}$ | $1.25 \cdot 10^{-4}$ |
| Performer | $1.2 \cdot 10^{-4}$ | $1.0 \cdot 10^{-4}$ |

Table 4: Best Learning Rate values obtained from 20 runs of Bayesian hyperparameters search

## B   PROPOSITION PROOFS

### B.1   CIRCULANT MATRICES

To design a more efficient positional encoding method, we leveraged circulant matrices – a subclass of matrices with some special properties due to their relation to the Fast Fourier Transform (FFT) and circular convolution Bamieh (2020). Here we will only focus on the property that allows performing matrix-vector product fast and efficient in terms of speed and memory. Given an vector $c = (c_0, c_1, ..., c_{n-1})$ we will define the associated $n \times n$ circulant matrix $C = \text{circ}(c)$ which first column is exactly $c$, and each subsequent column is obtained by a circular shift of the previous column:

$$C = \begin{pmatrix} c_0 & c_{n-1} & c_{n-2} & \cdots & c_1 \\ c_1 & c_0 & c_{n-1} & & c_2 \\ c_2 & c_1 & c_0 & & c_3 \\ \vdots & & \ddots & \ddots & \vdots \\ c_{n-1} & c_{n-2} & c_{n-3} & \cdots & \end{pmatrix} \tag{13}$$

It can be shown that for every vector $x$ of size $n$ matrix-vector product $Cx$ requires only $\mathcal{O}(n \log n)$ computation Rosowski (2021). Moreover, to compute the above product, it is not necessary to store the whole matrix $C$ in memory; it is enough to only keep in memory $\mathcal{O}(n)$ parameters of vector $c$. Further, we will prove that FastRPB and FastRPB 2D can be expressed through circular matrices, and hence relative positional information can be embedded efficiently in the self-attention mechanism.

### B.2   PROPOSITION 1

In 3.2.1 we introduced a Toeplitz matrix $W_{1d}$ of shape $N \times N$:

$$W_{1d} = \begin{pmatrix} w_0 & w_1 & w_2 & \cdots & w_{N-1} \\ w_{-1} & w_0 & w_1 & \cdots & w_{N-2} \\ w_{-2} & w_{-1} & w_0 & & w_{N-3} \\ \vdots & & \ddots & \ddots & \vdots \\ w_{-N+1} & w_{-N+2} & w_{-N+3} & \cdots & w_0 \end{pmatrix} \tag{14}$$

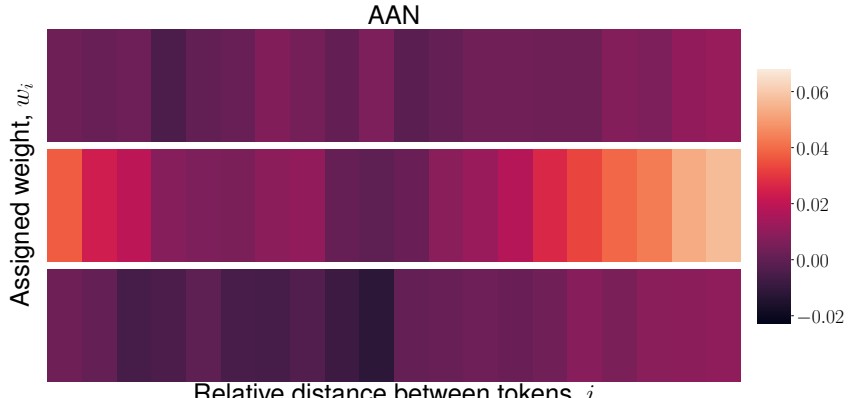

(a) Learned in AAN task weights $w_i$ assigned to pairwise distances between tokens $i$ in FastRPB 1D.

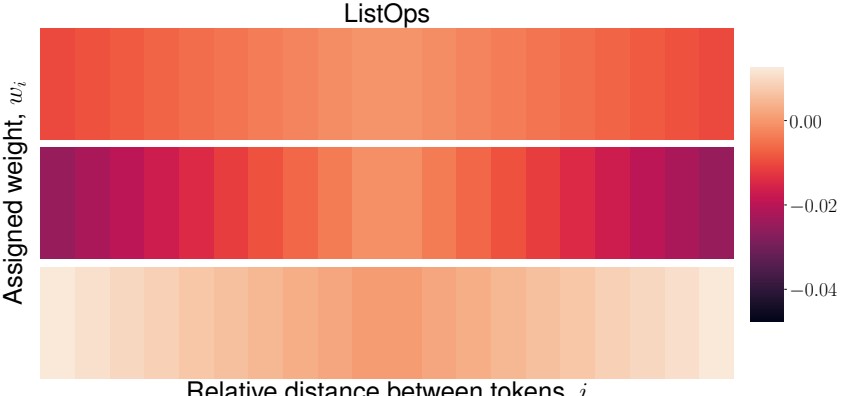

(b) Learned in ListOps task weights $w_i$ assigned to pairwise distances between tokens $i$ in FastRPB 1D.

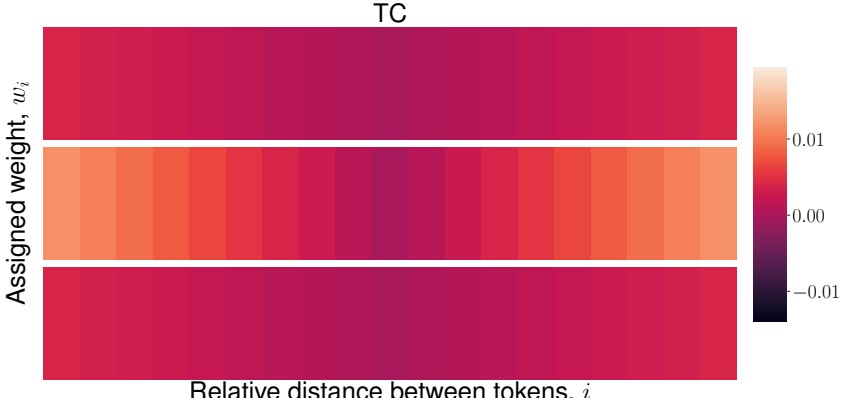

(c) Learned in TC task weights $w_i$ assigned to pairwise distances between tokens $i$ in FastRPB 1D.

Our goal is to efficiently multiply $\boldsymbol{W}_{1d}$ by arbitrary matrix $\boldsymbol{V}$ of shape $N \times D$. As was stated in section Circulant matrices, a special class of matrices, namely circulant matrices, can be multiplied by a vector efficiently in $\mathcal{O}(N \log N)$ operations and requires $\mathcal{O}(N)$ memory. We will extend matrix $\boldsymbol{W}_{1d}$ with additional rows and columns and thus obtain circulant matrix $\boldsymbol{W}_{1d}^{\text{ext}}$. Then we will introduce $\boldsymbol{V}^{\text{ext}}$, a modified version of matrix $\boldsymbol{V}$, which $\boldsymbol{W}_{1d}^{\text{ext}}$ will be multiplied by. Finally, we will select a slice from the product $\boldsymbol{W}_{1d}^{\text{ext}} \cdot \boldsymbol{V}^{\text{ext}}$, which will be exactly $\boldsymbol{W}_{1d} \cdot \boldsymbol{V}$.

The first step is to define $\boldsymbol{W}_{1d}^{\text{ext}}$:

$$\boldsymbol{W}_{1d}^{\text{ext}} = \begin{pmatrix} w_{-N+1} & w_{-N+2} & \cdots & w_0 & w_1 & w_2 & \cdots & w_{N-1} \\ w_{N-1} & w_{-N+1} & \cdots & w_{-1} & w_0 & w_1 & \cdots & w_{N-2} \\ w_{N-2} & w_{N-1} & \cdots & w_{-2} & w_{-1} & w_0 & & w_{N-3} \\ \vdots & & \ddots & & & \ddots & \ddots & \vdots \\ w_1 & w_2 & \cdots & w_{-N+1} & w_{-N+2} & \cdots & w_{-1} & w_0 \\ \vdots & & \ddots & & & \ddots & \ddots & \vdots \\ w_{-N+2} & w_{-N+3} & \cdots & \cdots & \cdots & \cdots & w_{N-1} & w_{-N+1} \end{pmatrix} \quad (15)$$

As can be seen, constructed matrix $\boldsymbol{W}_{1d}^{\text{ext}}$ is indeed circulant. Moreover, the right upper corner of $\boldsymbol{W}_{1d}^{\text{ext}}$ is essentially $\boldsymbol{W}_{1d}$. Hence, $\boldsymbol{W}_{1d}$ can be expressed as a slice of $\boldsymbol{W}_{1d}^{\text{ext}}$ following numpy notation: $\boldsymbol{W}_{1d} = \left( \boldsymbol{W}_{1d}^{\text{ext}} \right)_{N:\,,\,:N+1}$.

Now we want to calculate matrix product $\boldsymbol{W}_{1d} \cdot \boldsymbol{V}$ using matrix $\boldsymbol{W}_{1d}^{\text{ext}}$ of size $(2N-1) \times (2N-1)$. Due to this fact we will need to multiply $\boldsymbol{W}_{1d}^{\text{ext}}$ with appropriate matrix $\boldsymbol{V}^{\text{ext}}$ of size $2N-1 \times D$. Since, as we seen, $\boldsymbol{W}_{1d}$ is a slice of $\boldsymbol{W}_{1d}^{\text{ext}}$, we only will need first $N$ rows of the resulting product $\boldsymbol{W}_{1d}^{\text{ext}} \cdot \boldsymbol{V}^{\text{ext}}$. More formally we need to find such matrix $\boldsymbol{V}^{\text{ext}}$ that: $\boldsymbol{W}_{1d} \cdot \boldsymbol{V} = \left( \boldsymbol{W}_{1d}^{\text{ext}} \cdot \boldsymbol{V}^{\text{ext}} \right)_{:N+1\,,\,:}$ To achieve with we will basically pad $\boldsymbol{V}$ with $N-1$ additional rows filled with zeros as follows:

$$\boldsymbol{V}^{\text{ext}} = \begin{pmatrix} 0 & 0 & \cdots & 0 \\ 0 & 0 & \cdots & 0 \\ \vdots & \vdots & \ddots & \vdots \\ 0 & 0 & \cdots & 0 \\ v_{0,0} & v_{0,1} & \cdots & v_{0,D-1} \\ v_{1,0} & v_{1,1} & \cdots & v_{1,D-1} \\ \vdots & \vdots & \ddots & \vdots \\ v_{N-1,0} & v_{N-1,1} & \cdots & v_{N-1,D-1} \end{pmatrix} \quad (16)$$

To complete the proof lets explicitly show that $\boldsymbol{W}_{1d} \cdot \boldsymbol{V} = \left( \boldsymbol{W}_{1d}^{\text{ext}} \cdot \boldsymbol{V}^{\text{ext}} \right)_{:N+1\,,\,:}$. Lets assume $D = 1$, generalization for bigger dimensions can be done using similar operations:

$$\left( \boldsymbol{W}_{1d}^{\text{ext}} \cdot \boldsymbol{V}^{\text{ext}} \right)_{:N+1\,,\,:} = \begin{pmatrix} w_{-N+1} & \cdots & w_0 & w_1 & \cdots & w_{N-1} \\ w_{N-1} & \cdots & w_{-1} & w_0 & \cdots & w_{N-2} \\ w_{N-2} & \cdots & w_{-2} & w_{-1} & & w_{N-3} \\ \vdots & \ddots & & & \ddots & \vdots \\ w_1 & \cdots & w_{-N+1} & w_{-N+2} & \cdots & w_0 \end{pmatrix} \begin{pmatrix} 0 \\ 0 \\ \vdots \\ 0 \\ v_0 \\ v_1 \\ \vdots \\ v_{N-1} \end{pmatrix} = \quad (17)$$

$$\begin{pmatrix} w_0 & w_1 & \cdots & w_{N-1} \\ w_{-1} & w_0 & \cdots & w_{N-2} \\ w_{-2} & w_{-1} & & w_{N-3} \\ \vdots & \ddots & \ddots & \vdots \\ w_{-N+1} & w_{-N+2} & \cdots & w_0 \end{pmatrix} \begin{pmatrix} v_0 \\ v_1 \\ \vdots \\ v_{N-1} \end{pmatrix} = \boldsymbol{W}_{1d} \cdot \boldsymbol{V} \quad (18)$$

The last thing we have to do is to calculate the complexity of matrix product of circulant matrix $\boldsymbol{W}_{1d}^{\text{ext}}$ with $\boldsymbol{V}^{\text{ext}}$. Since matrix $\boldsymbol{W}_{1d}^{\text{ext}}$ is circulant of size $(2N-1) \times (2N-1)$, according to the

section Circulant matrices, it requires $O(N)$ memory to it, and $\mathcal{O}(N \log N)$ operations to perform a matrix-vector product with vector if size $2N - 1$. In this regard to compute matrix product with matrix $V$ of size $N \times D$, we will need to perform $D$ times operations more, i.e. $\mathcal{O}(DN \log N)$ operations.

### B.3 PROPOSITION 2

In the following sections, we will be considering an example of $3 \times 3$ image. In section Structure of pairwise distances tensors we will study the general structure of tensors **X** and **Y**, which were introduced in section . In section Flattening of the tensors we will reshape these tensors and obtain a new pair of tensors $X_{\text{flat}}$ and $Y_{\text{flat}}$, which will be then efficiently multiplied by $V$ matrix in the final section Efficient matrix product.

#### B.3.1 STRUCTURE OF PAIRWISE DISTANCES TENSORS

First of all, to gain a deeper understanding of structure of tensors **X** and **Y**, we will explicitly write down components of this tensors for an image of size $3 \times 3$. Consider $X_{1,1,:,:}$ and $Y_{1,1,:,:}$ which contains weights assigned to vertical and horizontal relative distances from pixel $(1, 1)$ to all other pixels:

$$X_{1,1,:,:} = \begin{pmatrix} w_{-1} & w_{-1} & w_{-1} \\ w_0 & w_0 & w_0 \\ w_1 & w_1 & w_1 \end{pmatrix}, \ Y_{1,1,:,:} = \begin{pmatrix} w_{-1} & w_0 & w_1 \\ w_{-1} & w_0 & w_1 \\ w_{-1} & w_0 & w_1 \end{pmatrix} \tag{19}$$

One can observe, that $X_{n,m,:,:}$ and $Y_{n,m,:,:}$ have the property of symmetry, through which it can be proven that:

**Proposition B.1** $X_{n,i,:,:} = X_{n,j,:,:}$ and $Y_{i,n,:,:} = Y_{j,n,:,:}$ for every $n, i, j \in \{0, ..., N - 1\}$, and this property holds for image of every size.

Taking advantage of the proposition B.1 we can write out explicit form of tensors **X** and **Y** in case of $3 \times 3$ images. We will introduce the following notation:

$$A = \begin{pmatrix} w_{-2} & w_{-2} & w_{-2} \\ w_{-1} & w_{-1} & w_{-1} \\ w_0 & w_0 & w_0 \end{pmatrix}, \ B = \begin{pmatrix} w_{-1} & w_{-1} & w_{-1} \\ w_0 & w_0 & w_0 \\ w_1 & w_1 & w_1 \end{pmatrix}, \ C = \begin{pmatrix} w_0 & w_0 & w_0 \\ w_1 & w_1 & w_1 \\ w_2 & w_2 & w_2 \end{pmatrix} \tag{20}$$

It can be seen that different slices of tensor **X** can be expressed using matrices $A, B, C$:

$$X_{0,0,:,:} = X_{0,1,:,:} = X_{0,2,:,:} = C \tag{21}$$

$$X_{1,0,:,:} = X_{1,1,:,:} = X_{1,2,:,:} = B \tag{22}$$

$$X_{2,0,:,:} = X_{2,1,:,:} = X_{2,2,:,:} = A \tag{23}$$

Moreover, are matrices $A, B, C$ also applicable for tensor **Y**:

$$Y_{0,0,:,:} = Y_{0,1,:,:} = Y_{0,2,:,:} = C^T \tag{24}$$

$$Y_{1,0,:,:} = Y_{1,1,:,:} = Y_{1,2,:,:} = B^T \tag{25}$$

$$Y_{2,0,:,:} = Y_{2,1,:,:} = Y_{2,2,:,:} = A^T \tag{26}$$

#### B.3.2 FLATTENING OF THE TENSORS

In transformer architecture before processing the image of size $N \times N$, it is usually flattened into 1-dimensional vector of size $N^2$. To define flatten operation, consider an arbitrary matrix $M$ of shape $3 \times 3$, than its flattened version $m_{\text{flat}}$ will have the following structure:

$$M = \begin{pmatrix} m_{0,0} & m_{0,1} & m_{0,2} \\ m_{1,0} & m_{1,1} & m_{1,2} \\ m_{2,0} & m_{2,1} & m_{2,2} \end{pmatrix} \xrightarrow{\text{flattening}} m_{\text{flat}} = \begin{pmatrix} m_{0,0} \\ m_{0,1} \\ m_{0,2} \\ m_{1,0} \\ m_{1,1} \\ m_{1,2} \\ m_{2,0} \\ m_{2,1} \\ m_{2,2} \end{pmatrix} \tag{27}$$

Due to the flatten of images in transformer, matrix $V$ in will have shape $N^2 \times D$ and hence it is essential to reshape tensors **X** and **Y** from size $(N \times N) \times (N \times N)$ to size $N^2 \times N^2$. We will denoted reshaped versions of tensors **X** and **Y** as $X_{\text{flat}}$ and $Y_{\text{flat}}$ respectively. Reshaping of the above tensors can be decomposed into two consecutive flatten operations firstly applied to last two dims of tensors **X** and **Y** and then to the first two ones. Flattening of the last dimensions is equivalent to flattening of each of the matrix $A, B, C$, after this operation in case of $3 \times 3$ images we will obtain the following three vectors:

$$
\boldsymbol{a}_{\text{flat}} = \begin{pmatrix} w_0 \\ w_0 \\ w_0 \\ w_1 \\ w_1 \\ w_1 \\ w_2 \\ w_2 \\ w_2 \end{pmatrix} , \ \boldsymbol{b}_{\text{flat}} = \begin{pmatrix} w_{-1} \\ w_{-1} \\ w_{-1} \\ w_0 \\ w_0 \\ w_0 \\ w_1 \\ w_1 \\ w_1 \end{pmatrix} , \ \boldsymbol{c}_{\text{flat}} = \begin{pmatrix} w_{-2} \\ w_{-2} \\ w_{-2} \\ w_{-1} \\ w_{-1} \\ w_{-1} \\ w_0 \\ w_0 \\ w_0 \end{pmatrix}
\tag{28}
$$

Than after next flattening operation we will get:

$$
\boldsymbol{X}_{\text{flat}} = \textbf{X}.\text{reshape}(N^2, N^2) = \begin{pmatrix}
w_0 & w_0 & w_0 & w_1 & w_1 & w_1 & w_2 & w_2 & w_2 \\
w_0 & w_0 & w_0 & w_1 & w_1 & w_1 & w_2 & w_2 & w_2 \\
w_0 & w_0 & w_0 & w_1 & w_1 & w_1 & w_2 & w_2 & w_2 \\
w_{-1} & w_{-1} & w_{-1} & w_0 & w_0 & w_0 & w_1 & w_1 & w_1 \\
w_{-1} & w_{-1} & w_{-1} & w_0 & w_0 & w_0 & w_1 & w_1 & w_1 \\
w_{-1} & w_{-1} & w_{-1} & w_0 & w_0 & w_0 & w_1 & w_1 & w_1 \\
w_{-2} & w_{-2} & w_{-2} & w_{-1} & w_{-1} & w_{-1} & w_0 & w_0 & w_0 \\
w_{-2} & w_{-2} & w_{-2} & w_{-1} & w_{-1} & w_{-1} & w_0 & w_0 & w_0 \\
w_{-2} & w_{-2} & w_{-2} & w_{-1} & w_{-1} & w_{-1} & w_0 & w_0 & w_0
\end{pmatrix}
\tag{29}
$$

$$
\boldsymbol{Y}_{\text{flat}} = \textbf{Y}.\text{reshape}(N^2, N^2) = \begin{pmatrix}
w_0 & w_1 & w_2 & w_0 & w_1 & w_2 & w_0 & w_1 & w_2 \\
w_{-1} & w_0 & w_1 & w_{-1} & w_0 & w_1 & w_{-1} & w_0 & w_1 \\
w_{-2} & w_{-1} & w_0 & w_{-2} & w_{-1} & w_0 & w_{-2} & w_{-1} & w_0 \\
w_0 & w_1 & w_2 & w_0 & w_1 & w_2 & w_0 & w_1 & w_2 \\
w_{-1} & w_0 & w_1 & w_{-1} & w_0 & w_1 & w_{-1} & w_0 & w_1 \\
w_{-2} & w_{-1} & w_0 & w_{-2} & w_{-1} & w_0 & w_{-2} & w_{-1} & w_0 \\
w_0 & w_1 & w_2 & w_0 & w_1 & w_2 & w_0 & w_1 & w_2 \\
w_{-1} & w_0 & w_1 & w_{-1} & w_0 & w_1 & w_{-1} & w_0 & w_1 \\
w_{-2} & w_{-1} & w_0 & w_{-2} & w_{-1} & w_0 & w_{-2} & w_{-1} & w_0
\end{pmatrix}
\tag{30}
$$

Note that rows of the $X_{\text{flat}}$ and $Y_{\text{flat}}$ are $\boldsymbol{a}_{\text{flat}}, \boldsymbol{b}_{\text{flat}}$ and $\boldsymbol{c}_{\text{flat}}$.

Now we have flattened representation $X_{\text{flat}}$ and $Y_{\text{flat}}$ and the last step we have to take is to compute matrix product $X_{\text{flat}} \cdot V$ and $Y_{\text{flat}} \cdot V$.

### B.3.3 EFFICIENT MATRIX PRODUCT

**Calculation of $Y_{\text{flat}} \cdot V$**

In case of $3 \times 3$ images it is easy to see that the matrix $Y_{\text{flat}}$, presented in formula 30 consists of 9 identical blocks, we will denote this blocks as $\boldsymbol{R}$:

$$
\boldsymbol{Y}_{\text{flat}} = \begin{pmatrix} \boldsymbol{R} & \boldsymbol{R} & \boldsymbol{R} \\ \boldsymbol{R} & \boldsymbol{R} & \boldsymbol{R} \\ \boldsymbol{R} & \boldsymbol{R} & \boldsymbol{R} \end{pmatrix}, \text{ where } \boldsymbol{R} = \begin{pmatrix} w_0 & w_1 & w_2 \\ w_{-1} & w_0 & w_1 \\ w_{-2} & w_{-1} & w_0 \end{pmatrix}
\tag{31}
$$

Note that matrix $\boldsymbol{R}$ is just a block of $\boldsymbol{Y}$, not its element. Also a very important fact is that can matrix $\boldsymbol{R}$ is a Toeplitz, which means that it can be efficiently multiplied by a vector, according to Proposition 1. Our goal is to efficiently multiply $Y_{\text{flat}}$ with a matrix $V$ of shape $N^2 \times D$. Since $V$ is basically $D$ times stacked vectors of size $N^2$ it will be enough to consider matrix-vector product

of $\boldsymbol{Y}_{\text{flat}}$ with a vector $\boldsymbol{v}$ of shape $N^2$. The product can be expressed as:

$$
\boldsymbol{Y}_{\text{flat}} \cdot \boldsymbol{V} = \begin{pmatrix} \boldsymbol{R} & \boldsymbol{R} & \boldsymbol{R} \\ \boldsymbol{R} & \boldsymbol{R} & \boldsymbol{R} \\ \boldsymbol{R} & \boldsymbol{R} & \boldsymbol{R} \end{pmatrix} \begin{pmatrix} v_0 \\ v_1 \\ v_2 \\ v_3 \\ v_4 \\ v_5 \\ v_6 \\ v_7 \\ v_8 \end{pmatrix} = \begin{pmatrix} \boldsymbol{R} \cdot \begin{pmatrix} v_0 \\ v_1 \\ v_2 \end{pmatrix} + \boldsymbol{R} \cdot \begin{pmatrix} v_3 \\ v_4 \\ v_5 \end{pmatrix} + \boldsymbol{R} \cdot \begin{pmatrix} v_6 \\ v_7 \\ v_8 \end{pmatrix} \\ \vdots \\ \vdots \end{pmatrix} = \tag{32}
$$

$$
\begin{pmatrix} \boldsymbol{R} \cdot \begin{pmatrix} v_0 + v_3 + v_6 \\ v_1 + v_4 + v_7 \\ v_2 + v_5 + v_8 \end{pmatrix} \\ \vdots \\ \vdots \end{pmatrix} \tag{33}
$$

Lets introduce an additional notation, we will denote as $\boldsymbol{V}_m$ a tensor of size $N \times N \times D$, which was obtained as a reshape of matrix $\boldsymbol{V}$ of size $N^2 \times D$. $\boldsymbol{V}_m$ will have the following structure:

$$
\boldsymbol{V}_m = \begin{pmatrix} v_0 & v_1 & v_2 \\ v_3 & v_4 & v_5 \\ v_6 & v_7 & v_8 \end{pmatrix} = \boldsymbol{V}.\text{reshape(N, N, D)} \tag{34}
$$

Then:

$$
\boldsymbol{V}_m^T.\text{sum}(1) = \begin{pmatrix} v_0 + v_3 + v_6 \\ v_1 + v_4 + v_7 \\ v_2 + v_5 + v_8 \end{pmatrix} \tag{35}
$$

Finally using notation 34 and result of product 33, we can conclude that:

$$
\boldsymbol{Y}_{\text{flat}} \cdot \boldsymbol{V} = \begin{pmatrix} \boldsymbol{R} \cdot \left( \boldsymbol{V}_m^T.\text{sum}(1) \right) \\ \vdots \\ \vdots \end{pmatrix} \tag{36}
$$

Note that the summation by dim $= 1$, not by dim $= -1$, since in general case, when $D$ is not 1, the latter dimension refers to hidden size.

$\boldsymbol{R}$ is a Topleitz matrix we worked with in the previous paragraph B.2, that is, the product of matrix $\boldsymbol{R}$ with $\boldsymbol{V}_m^T.\text{sum}(1)$ can be efficiently computed using the properties of Toeplitz matrix. Finally we can conclude that for $N \times N$ image computation of product $\boldsymbol{Y}_{\text{flat}} \cdot \boldsymbol{V}$ require $\mathcal{O}(DN \log N)$ computations and $\mathcal{O}(N)$ memory to store a vector, generating matrix $\boldsymbol{R}$.

**Calculation of $\boldsymbol{X}_{\text{flat}} \cdot \boldsymbol{V}$**

Here we will apply the following trick: by rearranging the columns of matrix $\boldsymbol{X}_{\text{flat}}$ we will construct matrix $\boldsymbol{X}'_{\text{flat}}$, and accordingly construct matrix $\boldsymbol{U}$ by rearranging the rows of matrix $\boldsymbol{V}$ in such a way that the equality holds $\boldsymbol{X}_{\text{flat}} \cdot \boldsymbol{V} = \boldsymbol{X}'_{\text{flat}} \cdot \boldsymbol{U}$. We still will work with the $D = 1$ case since, as was noted previously, it can be easily generalized for higher dimensions.

$$
\boldsymbol{X}_{\text{flat}} \cdot \boldsymbol{V} = \begin{pmatrix} w_0 & w_0 & w_0 & w_1 & w_1 & w_1 & w_2 & w_2 & w_2 \\ w_0 & w_0 & w_0 & w_1 & w_1 & w_1 & w_2 & w_2 & w_2 \\ w_0 & w_0 & w_0 & w_1 & w_1 & w_1 & w_2 & w_2 & w_2 \\ w_{-1} & w_{-1} & w_{-1} & w_0 & w_0 & w_0 & w_1 & w_1 & w_1 \\ w_{-1} & w_{-1} & w_{-1} & w_0 & w_0 & w_0 & w_1 & w_1 & w_1 \\ w_{-1} & w_{-1} & w_{-1} & w_0 & w_0 & w_0 & w_1 & w_1 & w_1 \\ w_{-2} & w_{-2} & w_{-2} & w_{-1} & w_{-1} & w_{-1} & w_0 & w_0 & w_0 \\ w_{-2} & w_{-2} & w_{-2} & w_{-1} & w_{-1} & w_{-1} & w_0 & w_0 & w_0 \\ w_{-2} & w_{-2} & w_{-2} & w_{-1} & w_{-1} & w_{-1} & w_0 & w_0 & w_0 \end{pmatrix} \cdot \begin{pmatrix} v_0 \\ v_1 \\ v_2 \\ v_3 \\ v_4 \\ v_5 \\ v_6 \\ v_7 \\ v_8 \end{pmatrix} = \tag{37}
$$

$$
\begin{pmatrix}
w_0 & w_1 & w_2 & w_0 & w_1 & w_2 & w_0 & w_1 & w_2 \\
w_0 & w_1 & w_2 & w_0 & w_1 & w_2 & w_0 & w_1 & w_2 \\
w_0 & w_1 & w_2 & w_0 & w_1 & w_2 & w_0 & w_1 & w_2 \\
w_{-1} & w_0 & w_1 & w_{-1} & w_0 & w_1 & w_{-1} & w_0 & w_1 \\
w_{-1} & w_0 & w_1 & w_{-1} & w_0 & w_1 & w_{-1} & w_0 & w_1 \\
w_{-1} & w_0 & w_1 & w_{-1} & w_0 & w_1 & w_{-1} & w_0 & w_1 \\
w_{-2} & w_{-1} & w_0 & w_{-2} & w_{-1} & w_0 & w_{-2} & w_{-1} & w_0 \\
w_{-2} & w_{-1} & w_0 & w_{-2} & w_{-1} & w_0 & w_{-2} & w_{-1} & w_0 \\
w_{-2} & w_{-1} & w_0 & w_{-2} & w_{-1} & w_0 & w_{-2} & w_{-1} & w_0
\end{pmatrix}
\cdot
\begin{pmatrix}
v_0 \\ v_3 \\ v_6 \\ v_1 \\ v_4 \\ v_7 \\ v_2 \\ v_5 \\ v_8
\end{pmatrix}
= \boldsymbol{X}'_{\text{flat}} \cdot \boldsymbol{U} =
\begin{pmatrix}
\alpha \\ \alpha \\ \alpha \\ \beta \\ \beta \\ \beta \\ \gamma \\ \gamma \\ \gamma
\end{pmatrix}
\tag{38}
$$

Now lets use already introduced matrix $\boldsymbol{V}_m$:

$$
\boldsymbol{R} \cdot \left( \boldsymbol{V}_m.\text{sum}(1) \right) =
\begin{pmatrix}
w_0 & w_1 & w_2 \\
w_{-1} & w_0 & w_1 \\
w_{-2} & w_{-1} & w_0
\end{pmatrix}
\begin{pmatrix}
v_0 + v_1 + v_2 \\
v_3 + v_4 + v_5 \\
v_6 + v_7 + v_8
\end{pmatrix}
=
\tag{39}
$$

$$
\boldsymbol{R} \cdot
\begin{pmatrix} v_0 \\ v_3 \\ v_6 \end{pmatrix}
+ \boldsymbol{R} \cdot
\begin{pmatrix} v_1 \\ v_4 \\ v_7 \end{pmatrix}
+ \boldsymbol{R} \cdot
\begin{pmatrix} v_2 \\ v_5 \\ v_8 \end{pmatrix}
=
\begin{pmatrix} \alpha \\ \beta \\ \gamma \end{pmatrix}
\tag{40}
$$

And finally:

$$
\boldsymbol{X}'_{\text{flat}} \cdot \boldsymbol{U} =
\begin{pmatrix}
\alpha \\ \alpha \\ \alpha \\ \beta \\ \beta \\ \beta \\ \gamma \\ \gamma \\ \gamma
\end{pmatrix}
=
\begin{pmatrix}
\alpha & \alpha & \alpha \\
\beta & \beta & \beta \\
\gamma & \gamma & \gamma
\end{pmatrix}
.\text{reshape}(N^2) =
\tag{41}
$$

$$
\left( \boldsymbol{R} \cdot \left( \boldsymbol{V}_m.\text{sum}(1) \right) \quad \cdots \quad \cdots \right) .\text{reshape}(N^2)
\tag{42}
$$

### B.3.4 CONCLUSION

As we proved in the previous section, products $\boldsymbol{X}_{\text{flat}} \cdot \boldsymbol{V}$ and $\boldsymbol{Y}_{\text{flat}} \cdot \boldsymbol{V}$ can be computed using just two products: $\boldsymbol{R} \cdot \left( \boldsymbol{V}_m.\text{sum}(1) \right)$ and $\boldsymbol{R} \cdot \left( \boldsymbol{V}_m^T.\text{sum}(1) \right)$ respectively. Moreover, matrix $\boldsymbol{R}$ utilized in the above products is a Toeplitz, and in this regard this products can be computed efficiently according to Proposition 1. Summarizing it all we get that FastRBP 2D will require be $\mathcal{O}(DN \log N)$ computations and $\mathcal{O}(N)$ memory.

