# OpenReview forum: "FastRPB: a Scalable Relative Positional Encoding for Long Sequence Tasks"
_ICLR.cc/2022/Conference — ICLR 2022 Submitted_

### Official Review · Reviewer_7TLj · 2021-10-30

**Correctness:** 3
**Technical Novelty And Significance:** 3
**Empirical Novelty And Significance:** 2
**Recommendation:** 5
**Confidence:** 2

**Main Review:**

Strengths:

1. The proposed FastRPB gained a higher training speed and required less memory.

Weaknesses:

1. This is an incremental study for two existing issues of the Transformer framework on the long sequence tasks.

2. The proposed approach does not achieve good results on all tasks in Tables 1 and 3.

Comments:

1. The whole paper indeed explored two existing issues of the Transformer framework: inefficient scaling and weak performance on long sequence tasks. However, this title only focused on the first issue: inefficient scaling on long sequence tasks. What is the relationship between inefficient scaling and weak performance on long sequence tasks?

2. Experimental results demonstrated that the proposed two methods cooperated with each other to improve the performance of Linear Transformer models on part of long sequence tasks. What kind of information is captured by the proposed two methods to improve the performance of Linear Transformer models on part of long sequence tasks? Particularly, both of them cooperated with each other.

**Summary Of The Paper:**

This paper focused on two existing issues of the Transformer framework: inefficient scaling and weak performance on long sequence tasks. Take the Linear Transformer model as an example, the authors proposed the Shift-Invariant Kernel Function (SIKF) for the weak performance issue and the Fast Relative Positional Bias (FastRPB) for the weak performance issue, respectively. The proposed methods were evaluated on several long sequence tasks.

**Summary Of The Review:**

This is an incremental study for two existing issues of the Transformer framework on the long sequence tasks.

---

> ### Author Response · Authors · 2021-11-21
> **On the review**
>
> Thank you for the detailed review! We will take into account the recommendations left by you, make conclusions, and do not make mistakes in the future

---

### Official Review · Reviewer_8WRr · 2021-11-03

**Correctness:** 3
**Technical Novelty And Significance:** 2
**Empirical Novelty And Significance:** 2
**Recommendation:** 5
**Confidence:** 4

**Main Review:**

Strengths:
1. The proposed method can solve the limitations of Linear Transformer by improving the kernel function and relative position bias.
2. The proposed method show improvement over linear Transformer on several benchmark dataset.

Weakness:
1. The proposed method is still not comparable to SOTA models on this benchmark, such as BigBird which is also a simple and effective method.
2. The modification over Linear Transformer is limited, especially the kernel part. Thus it's not surprising that the speed and memory usage is close to Linear Transformer.
3. FastRPB is quite a general method. It would be better to test it with different Transformer structures.
4. FastRPB is more novel, but it cannot lead to better performance than other baselines in most cases.
5. Related work is too long.
6. It would be better if the proposed method can further improve the vision Transformer on imagenet.

**Summary Of The Paper:**

The proposed method is mainly based on Linear Transformer. The authors propose Shift-Invariant Kernel Function (SIKF) by replacing the kernel function in Linear Transformer with exp (x), which satisfies the property of shift-invariance. Besides, the authors also proposed FastRPB which adds Relative Positional Bias to self-attention weights. The experiments show that both methods are important to improve Linear Transformer. Moreover, the proposed method is fast and memory efficient.

**Summary Of The Review:**

Overall, the improvement over kernel function and position bias is simple, but it hasn't led to significant improvement.  The performance is still far behind SOTA performance. Either SIKF or FastRPB is actually not leading to significant improvement over baselines on most of the tasks.

---

> ### Author Response · Authors · 2021-11-21
> **On the review**
>
> Thank you for the detailed review! We will take into account the recommendations left by you, make conclusions, and do not make mistakes in the future

---

### Official Review · Reviewer_VGvT · 2021-11-03

**Correctness:** 2
**Technical Novelty And Significance:** 2
**Empirical Novelty And Significance:** 2
**Recommendation:** 3
**Confidence:** 4

**Main Review:**

The main contributions of the papers are: 1) a new kernel for Linear Transformer using the exponential function as the activation and 2) a fast way to apply relative positional bias to attention, in both 1-d and 2-d sequence cases.

The paper claims that the shift-invariant property of the exponential function can improve the performance of linear Transformers, while the experimental results do not support this claim. From Table 1, most of the best models are with DPFP or ReLU activations. Besides, the paper doesn't present more theoretical justification for their choice of the kernel function.

Besides, the paper missed an important citation [1]. [1] described the exact same method as the fast relative positional bias proposed in this paper, i.e., using FFT to accelerate multiplying a Toeplitz positional bias matrix in the 1D sequence setting. But this paper further generalizes the fast relative positional bias to the 2D sequence case to deal with image-like data, which can be regarded as a new contribution.

Finally, the paper conduct experimental results on the long-range-arena benchmark, where most tasks are of very small scales. I would suggest the authors further conduct one or two large-scale experiments (e.g., GLUE, machine translation, language modeling, ImageNet classification) to verify the effectiveness of the proposed model.

[1]: Stable, Fast and Accurate: Kernelized Attention with Relative Positional Encoding. NeurIPS 2021

**Summary Of The Paper:**

The paper proposes two modifications of linear Transformer: 1) a new kernel for Linear Transformer using the exponential function as the activation and 2) a fast way to apply relative positional bias to attention, in both 1-d and 2-d sequence cases. The experiments are only conducted on the long-range-arena benchmark, and the performance improvement does not seem to be significant.


**Summary Of The Review:**

Given the weak theoretical and experimental supports of the proposed method and duplication of previous work, I would recommend the rejection of the paper.

---

> ### Author Response · Authors · 2021-11-21
> **On the review**
>
> Thank you for the detailed review! We will take into account the recommendations left by you, make conclusions, and do not make mistakes in the future

---

### Official Review · Reviewer_L9JY · 2021-11-04

**Correctness:** 3
**Technical Novelty And Significance:** 2
**Empirical Novelty And Significance:** 2
**Recommendation:** 5
**Confidence:** 3

**Main Review:**

Strengths:
1. The proposed SIKF kernel is fast in practice, easy in implementation and achieves good performance
2. The proposed Relative Position Bias method provides a novel method of incorporating relative positional bias for Linear Transformers. The presented experimental results show that the proposed method outperforms the baselines for AAN and TC.

Weaknesses :

1. Regarding Shift Invariance 1: The authors claim that the shift invariant property of the softmax function is important for Transformers performance. However, why this might be important is not very well motivated: what causes this property to help with the performance?

2. Regarding Shift Invariance 2: The original \hat{SM_{m}}^{+} estimator presented in the Performer paper is also shift invariant. From the current version of the paper, it is not very clear why having SIKF would be more advantageous, since both offer shift invariance.

3. Regarding improved performance of SIKF: The claim that the SIKF kernel offers comparable performance to existing methods is slightly strong. From Table 1, considering the performance without any positional embeddings to remove confounders, the Linear DPFP kernel outperforms SIKF on all 4 tasks (noteworthy being ListOps with an 8 point delta). Additionally, from Table 3, the peak memory usage as well as training time for Linear DPFP and SIKF is very similar for the no positional encoding setting. Thus the benefits of SIKF are somewhat unclear.

4. Regarding stability of SIKF: Computing dot product of exponentiated quantities might result in overflow issues, especially when training with reduced precision. It would be good to have an experiment to show if that is indeed the case for the SIKF kernel or not.

5. Regarding FastRPB: The proposal for using scalars for distances between two points is quite commonly used (eg [1]). However, a key difference between the previous approaches and the one presented in this paper is that of binning the large distances (done logarithmically in [1]). Having that variant in the experiments would be quite useful (especially for tasks where the sequence lengths vary and hence the updates to the larger difference weights is a lot more sparse).

6. CIFAR Results: The reason given for the poor result on CIFAR is that only a single layer network was utilised. However, it is not very clear why that would result in a poor performance for FastRPB. What property of small networks affects the generalisability of FastRPB, and why does that not factor in for the other positional encoding methods.

7. Results presented on MNIST: Given that MNIST is a fairly easy dataset (eg: 2 layer MLPs achieve strong accuracy on the dataset), training a 8 layer Transformer for the task might not be necessary. In my opinion, if the purpose of the experiment is to show generalisation on a larger Vision dataset, where training a larger model is necessary, then considering CIFAR 100 / mini Imagenet might be better to draw conclusions.

8. Besides training time, reporting inference time (eg: on dev set for all the models) might also help compare model speed: a model may be slower to train, but might have faster inference time. This would help distinguish between that.


Presentation improvement suggestions:
1. For Equation (2) and (3), having the summation outside the fraction would help improve clarity
2. Section 3.2, last para: we can pad the "shorter" one to make the input and output lengths match.

References

[1] Raffel, Colin, et al. "Exploring the limits of transfer learning with a unified text-to-text transformer." arXiv preprint arXiv:1910.10683 (2019).


**Summary Of The Paper:**

The paper presents two contributions
1. It expands on Linear Transformers by introducing a feature map \phi = exp(x) function whose dot product approximates the softmax kernel (dubbed Shift Invariant Kernel Function SIKF); whose motivation comes from the shift invariant property of the softmax function. They show that it performs comparably to other proposed approximations in literature while improving training efficiency and reducing the memory footprint
2. It presents a way to use relative position bias with Linear Transformers. Dubbed Fast Relative Position Bias (FastRPB), the method proposes to learn a scalar for pairwise difference of positions, and draw a connection between the distance matrix and Circulant Matrices. They use this to efficiently compute the relative position bias in O(ND log N ) operations with O(N) memory. They show that FastRBP outperforms other relative positional bias methods while not substantially increasing compute.

**Summary Of The Review:**

The paper presents a fast novel kernel (SIKF) and relative positional embeddings for Linear Transformers (FastRPB). The results with SIKF are quite mixed, and the motivation not very clear. FastRPB shows good gains against other baselines on 2 of the 4 datasets presented. However for the other two, the results are quite mixed. The reasoning provided for why FastRPB is bad needs further investigation. Additionally, having results on an image dataset more difficult than MNIST would help draw better conclusions.

---

> ### Author Response · Authors · 2021-11-21
> **On the review**
>
> Thank you for the detailed review! We will take into account the recommendations left by you, make conclusions, and do not make mistakes in the future

---

### Decision · Program_Chairs · 2022-01-20

**Decision:**

Reject

**Comment:**

All reviewers are in agreement to reject this paper. The main objection is that the tasks chosen are small scale and that the mixed results are not strong enough. The authors did not attempt to raise substantial issues to be discussed.